# The Phenoxyphenol Compound diTFPP Mediates Exogenous C_2_-Ceramide Metabolism, Inducing Cell Apoptosis Accompanied by ROS Formation and Autophagy in Hepatocellular Carcinoma Cells

**DOI:** 10.3390/antiox10030394

**Published:** 2021-03-05

**Authors:** Wen-Tsan Chang, Yung-Ding Bow, Yen-Chun Chen, Chia-Yang Li, Jeff Yi-Fu Chen, Yi-Ching Chu, Yen-Ni Teng, Ruei-Nian Li, Chien-Chih Chiu

**Affiliations:** 1Department of Surgery, Division of General and Digestive Surgery, Kaohsiung Medical University Hospital, Kaohsiung 807, Taiwan; wtchang@kmu.edu.tw; 2Department of Surgery, School of Medicine, College of Medicine, Kaohsiung Medical University, Kaohsiung 807, Taiwan; 3Center for Cancer Research, Kaohsiung Medical University Hospital, Kaohsiung Medical University, Kaohsiung 807, Taiwan; 4College of Life Science, Kaohsiung Medical University; Kaohsiung 807, Taiwan; u109850001@kmu.edu.tw; 5Department of Biotechnology, Kaohsiung Medical University, Kaohsiung 807, Taiwan; r020135@gap.kmu.edu.tw (Y.-C.C.); yifuc@kmu.edu.tw (J.Y.-F.C.); 6Graduate Institute of Medicine, College of Medicine, Kaohsiung Medical University, Kaohsiung 807, Taiwan; chiayangli@kmu.edu.tw; 7Department of Medicinal and Applied Chemistry, Kaohsiung Medical University, Kaohsiung 807, Taiwan; N56094512@ncku.edu.tw; 8Department of Biological Sciences and Technology, National University of Tainan, Tainan 700, Taiwan; tengyenni@mail.nutn.edu.tw; 9Department of Biomedical Science and Environment Biology, Kaohsiung Medical University, Kaohsiung 807, Taiwan; runili@kmu.edu.tw; 10Department of Biological Sciences, National Sun Yat-Sen University, Kaohsiung 804, Taiwan; 11The Graduate Institute of Medicine, Kaohsiung Medical University, Kaohsiung 807, Taiwan; 12Department of Medical Research, Kaohsiung Medical University Hospital, Kaohsiung 807, Taiwan

**Keywords:** hepatocellular carcinoma (HCC), phenoxyphenol compound, diTFPP, ROS, apoptosis, autophagy

## Abstract

Hepatocellular carcinoma (HCC) is a severe disease that accounts for 80% of liver cancers. Chemotherapy is the primary therapeutic strategy for patients who cannot be treated with surgery or who have late-stage HCC. C_2_-ceramide is an effective reagent that has been found to inhibit the growth of many cancer types. The metabolism of C_2_-ceramide plays a vital role in the regulation of cell death/cell survival. The phenoxyphenol compound 4-{2,3,5,6-tetrafluoro-4-[2,3,5,6-tetrafluoro-4-(4-hydroxyphenoxy)phenyl]phenoxy}phenol (diTFPP) was found to have a synergistic effect with C_2_-ceramide, resulting in considerable cell death in the HA22T HCC cell line. diTFPP/C_2_-ceramide cotreatment induced a two- to threefold increase in cell death compared to that with C2-ceramide alone and induced pyknosis. Annexin V/7-aminoactinomycin D (7AAD) double staining and Western blotting indicated that apoptosis was involved in diTFPP/C_2_-ceramide cotreatment-mediated cell death. We next analyzed transcriptome alterations in diTFPP/C_2_-ceramide-cotreated HA22T cells with next-generation sequencing (NGS). The data indicated that diTFPP treatment disrupted sphingolipid metabolism, inhibited cell cycle-associated gene expression, and induced autophagy and reactive oxygen species (ROS)-responsive changes in gene expression. Additionally, we assessed the activation of autophagy with acridine orange (AO) staining and observed alterations in the expression of the autophagic proteins LC3B-II and Beclin-1, which indicated autophagy activation after diTFPP/C_2_-ceramide cotreatment. Elevated levels of ROS were also reported in diTFPP/C_2_-ceramide-treated cells, and the expression of the ROS-associated proteins SOD1, SOD2, and catalase was upregulated after diTFPP/C_2_-ceramide treatment. This study revealed the potential regulatory mechanism of the novel compound diTFPP in sphingolipid metabolism by showing that it disrupts ceramide metabolism and apoptotic sphingolipid accumulation.

## 1. Introduction

Liver cancer is the fourth most common cause of cancer-related death worldwide, resulting in over 700000 deaths in 2018 [1,2]. Due to the development of medical science, the mortality rates of other major cancers, namely, prostate, breast, and colorectal cancer, have declined, but the mortality rate of liver cancer has rapidly increased [1,3]. Hepatocellular carcinoma (HCC) is derived from hepatocytes, which account for approximately 80% of primary liver cancers [4]. Chemotherapy is one of the most critical therapeutic procedures against advanced HCC, and it acts by inducing programmed cell death (PCD), especially apoptosis [5,6]. However, many HCCs are chemotherapy resistant [7]. Therefore, resensitizing cancer cells to chemotherapy drugs is a potential strategy for the development of chemotherapy.

Ceramide is a sphingolipid and structural molecule of the cell membrane that regulates fluidity [8]. Ceramide was first reported to induce cell apoptosis in leukemia 30 years ago [9]. The anticancer potential of ceramide has been found against many cancer types, such as nonsmall-cell lung carcinoma [10,11], head and neck squamous cell carcinoma [12], breast cancer [13], and multiple myeloma [14]. The metabolism of sphingolipids regulates the fate of the cell. Sphingolipids are roughly divided into prosurvival sphingolipids and apoptotic sphingolipids [15]. Sphingosine and ceramide are considered apoptotic sphingolipids, inducing cell death by regulating the apoptotic pathway, including extrinsic and intrinsic pathways [16]. The metabolites of ceramide and sphingosine, such as ceramide-1-phosphate (C1P), sphingosine-1-phosphate (S1P), or glucosylceramide, are considered prosurvival sphingolipids that prevent the cell from undergoing apoptosis [17,18,19]. Ceramides have been reported to play an essential role in the crosstalk of protective autophagy and apoptotic autophagy [15].

Autophagy is a catabolic procedure that degrades biological waste, misfolded proteins, or damaged organelles [20]. As indicated in our previous studies, autophagy plays a role in many anticancer drug treatments and leads to cell death or cell survival [10,21,22,23]. As a double-edged sword for apoptosis, autophagy blocks the caspase cascade or removes damaged organelles, releasing apoptotic signals to prevent apoptosis [24,25], but in some cases, autophagy acts as the caspase activation platform triggering apoptosis [26,27]. Reactive oxidative species (ROS) are free radical or nonradical oxygen species, including superoxide anion and hydrogen peroxide, that lead to oxidative stress and many diseases. ROS are also involved in autophagy, apoptosis, and apoptotic cell death regulation. In our previous study, ROS were shown to be associated with chemotherapeutic drug treatment and apoptosis.

4-{2,3,5,6-Tetrafluoro-4-[2,3,5,6-tetrafluoro-4-(4-hydroxyphenoxy)phenylsphenoxy} phenol (diTFPP) is a kind of phenoxyphenol that contains 1 more tetrafluorobenzene than 4-[2,3,5,6-tetrafluoro-4-(4-hydroxyphenoxy)phenoxy]phenol (TFPP, Figure 1A). It has been shown to exert a synergistic effect with camptothecin (CPT) and induce apoptosis [28] In our previous study, phenoxyphenol compounds were observed to either induce cell apoptosis [29] or sensitize cells to chemotherapeutic agents [28]. Therefore, in this study, we revealed the role of diTFPP in sensitizing HCC to C_2_-ceramide by activating the ROS/autophagy pathway.

## 2. Materials and Methods

### 2.1. Cell Culture

HA22T/VGH (HA22T, #60168) cells were purchased from the Bioresource Collection and Research Center (BCRC, Hsinchu, Taiwan) and maintained in a 3:2 mixture of Dulbecco’s modified Eagle medium and Ham’s F-12 Nutrient Mixture (DMEM/F12, 3:2; HIMEDIA, Mumbai, India) supplemented with 8% fetal bovine serum (FBS; ThermoFisher, Waltham, MA, USA) and 1% penicillin-streptomycin-amphotericin B (P/S/A; #03-033-1B, Biological Industries, Beit-Haemek, Israel). Cultures were grown in a 37 °C incubator with an atmosphere containing 5% CO_2_. Hematoxylin staining was utilized to visualize the cell nucleus. The treated cells were fixed with 4% paraformaldehyde (PFA) for 10 min and stained with hematoxylin (#GHS3, Sigma-Aldrich, St. Louis, MO, USA) for 5 min.

### 2.2. Cell Viability

The viability of HA22T cells was measured by trypan blue exclusion assay [23]. Briefly, the cells were resuspended in 0.05% trypsin (#TCL034, HIMEDIA, Mumbai, India) and exposed to 0.2% trypan blue reagent. Trypan blue dye did not stain the viable cells. Then, the viable cells were counted with a hemocytometer.

### 2.3. Measurement of Apoptotic Cells

Apoptotic HA22T cells were assessed by annexin V/7AAD double staining. An apoptosis detection kit (Strong Biotech Corporation, Taipei, Taiwan) and 7AAD (#11397, Cayman Chemicals, Ann Arbor, MI, USA) were used for annexin V/7AAD staining, and we replaced propidium iodide with 7AAD reagent. The procedure was performed according to the manufacturer’s instructions. Briefly, the treated cells were harvested and stained with annexin V/7AAD, analyzed with an LSR II flow cytometer (BD Biosciences, San Jose, CA, USA) and visualized with FlowJo 7.6.1 software (TreeStar, Inc., Ashland, OR, USA).

### 2.4. Western Blotting

Protein expression was measured by Western blot analysis. Briefly, the treated cells were harvested and lysed with lysis buffer, and the protein concentration was measured with a bicinchoninic acid (BCA) protein assay kit (Pierce, Rockford, IL, USA). An equal amount of protein (30 μg) was separated by SDS-polyacrylamide gel electrophoresis (SDS-PAGE) and electrotransferred to polyvinylidene difluoride (PVDF) membranes (Merck Millipore Ltd., Burlington, MA, USA) for one hour. The membranes were blocked with 5% nonfat milk in TBS buffer containing 0.1% Tween 20 (TBS-T buffer) and incubated overnight with primary antibodies targeting Bax (#50599-2-ig, Proteintech, Wuhan, Hubei, China), caspase-9 (#9508S, Cell Signaling Technology, Danvers, MA, USA), caspase-8 (#IR99-409, IReal Biotechnology, Hsinchu, Taiwan), PARP-1 (#SC-8007, Santa Cruz Biotechnology, Dallas, TX, USA), LC3B (#2775S, Cell Signaling Technology, Danvers, MA, USA), Beclin-1 (#3738, Cell Signaling Technology), SOD2 (#06-984, Merck, Darmstadt, Germany), β-actin (Sc-47778, Santa Cruz, Dallas, TX, USA), or glyceraldehyde-3-phosphate (GAPDH, #MAB374, EMD Millipore, Billerica MA, USA). Horseradish peroxidase (HRP)-conjugated secondary antibodies were then hybridized with the membrane for 1 h, and HRP activity was detected with an enhanced chemiluminescence (ECL) detection kit (PerkinElmer Inc, Waltham, MA, USA).

### 2.5. Next-Generation Sequencing Analysis

RNA library construction and sequencing were commissioned by Tools Biotech (BIOTOOLS, Taipei, Taiwan). The mRNAs of HCC cells will be randomly fragmented in a fragmentation buffer, and then cDNA synthesis will be performed using random hexamers and reverse transcriptase. After the first strand synthesis, a custom second strand synthesis buffer (Illumina, San Diego, CA, USA) and dNTPs, RNase H, and *E. coli* polymerase I were added to form a second strand. After purification, repair of the terminal, A-tailing, sequence adaptor ligation, size selection, and PCR enrichment, the final cDNA library was prepared for completion. Next, the cDNA size will be checked in the library adaptors at both ends and quantified to higher accuracy (library activity >2 nM) by quantitative PCR (Q-PCR). RNA libraries were sequenced, and the sequencing data were processed. The NGS data were clustered using Expander 7 software [30], and GO analysis was performed with the Database for Annotation, Visualization, and Integrated Discovery (DAVID) website (v6.8) [31]. The PCA and heat map were generated with the ClustVis website [32]. Gene set enrichment analysis (GSEA) was utilized to analyze the biological process alterations in the C_2_-ceramide and C_2_-ceramide/diTFPP groups. The gene set was obtained based on false discovery rate (FDR) <0.25 and *p* < 0.05.

### 2.6. AVO Staining

The assessment of autophagy will use flow cytometry-based acidic vesicular organelle (AVO) staining. Briefly, the cells were grown in 6-well plates at a density of 5 × 10^4^ cells per well, cultured for 24 h, and then treated with the indicated concentration of diTFPP (from 5 to 10 μM) combined with C_2_-ceramide for 24 h. The cells were collected and then stained with 1 μg/ml acridine orange (AO) at room temperature for 15 min. After the staining solution was removed, the cells were washed with phosphate-buffered saline (PBS) and immediately analyzed in an LSR II flow cytometer using 488-nm bandpass blue excitation filters and 515 nm (green) and 650 nm (red) barrier filters supported by the Center for Research Resources and Development of Kaohsiung Medical University, Taiwan.

### 2.7. ROS Detection

DHE was utilized to detect intracellular ROS formation. The cells were incubated with 1 μM DHE for 20 min and washed with PBS after incubation. The stained cells were visualized with an inverted fluorescence microscope.

### 2.8. Statistics

The comparison of two different groups was analyzed at least in triplicate by one-way analysis of variance (ANOVA), and the comparison between pairs was analyzed by Student’s *t*-test. A *p*-value < 0.05 was considered statistically significant.

## 3. Results

### 3.1. diTFPP Sensitizes Hepatocellular Carcinoma Cells to C_2_-Ceramide

C_2_-Ceramide is a ceramide with a methyl group on the R chain, and it contributes to apoptosis in cancer cells [10,11,12,33]. The human hepatocellular carcinoma cell line HA22T was observed to have resistance to C_2_-ceramide, which led to a cell death rate of less than 35% at a 20 μM dose (Figure 1B). Treatment with the phenoxyphenol compound diTFPP and 20 μM C_2_-ceramide led to a 70% cell death rate with no toxicity when diTFPP was used alone (Figure 1B). Cell morphology was also observed; treatment with diTFPP induced considerable cell death when coadministered with 20 μM C_2_-ceramide (Figure 1C). Next, we stained the cells with hematoxylin and observed pyknosis. Interestingly, the number of pyknotic cells increased after cotreatment with diTFPP/C_2_-ceramide (Appendix A).

### 3.2. The diTFPP/C_2_-Ceramide Cotreatment Triggers Apoptosis

Apoptosis is a well-known intercellular process associated with homeostasis, autophagy, and anticancer mechanisms, resulting in programmed cell death (PCD) [34,35]. Apoptosis is related to many chemotherapy drugs, such as irinotecan and fluorouracil (5-FU) [36,37,38]. To determine whether apoptosis is associated with the role of diTFPP/C_2_-ceramide-induced cell death, we analyzed the cells by annexin V/7-aminoactinomycin D (7AAD) double staining by flow cytometry. The cells in quadrants I (Q_I_) and IV (Q_IV_) were considered early-stage and late-stage apoptotic cells, respectively. The data indicated that cotreatment with diTFPP and C_2_-ceramide increased the percentage of annexin V-positive cells, suggesting that the synergism of diTFPP and C_2_-ceramide amplified apoptotic activity in HA22T cells (Figure 2A,B).

On the other hand, we also measured the expression alteration of apoptotic proteins by Western blot analysis. The extrinsic apoptotic protein caspase-8 was cleaved after diTFPP/C_2_-ceramide treatment but not in the other treatment groups. (Figure 2C). In addition, diTFPP/C_2_-ceramide cotreatment also induced the expression of caspase 8 downstream and the pro-apoptotic protein Bax as well as the cleavage of caspase-9 (Figure 2C). Poly (ADP-ribose) polymerase-1 (PARP-1) is associated with DNA repair, rescuing cells from cell damage. During the activation of apoptosis, PARP-1 is cleaved and deactivated, resulting in DNA double strain breakage and apoptosis [39]. Cleaved PARP-1 was upregulated significantly after diTFPP/C_2_-ceramide cotreatment (Figure 2C). Alterations in the percentage of annexin V-positive cells and the levels of proapoptotic protein markers indicated that the potential synergism of diTFPP and C_2_-ceramide plays a vital role in the activation of apoptosis.

### 3.3. Transcriptomic Analysis Reveals the Role of diTFPP/C_2_-Ceramide Treatment in Hepatocellular Carcinoma

Apoptosis was found to play a critical role in diTFPP/C_2_-ceramide-induced cell death, but the apoptotic mechanism of diTFPP/C_2_-ceramide is still unknown. Thus, we investigated alterations in the transcriptome after diTFPP/C_2_-ceramide treatment by next-generation sequencing (NGS). The results showed that the expression of over 20,000 genes was altered after 24 h of treatment with diTFPP, C_2_-ceramide, or diTFPP/C_2_-ceramide. Principal component analysis (PCA) was utilized to analyze the similarity of the transcriptome after the treatments. The cotreatment with the diTFPP/C_2_-ceramide group was found to be an immense distance from the other groups, suggesting that the cotreatment resulted in considerable alterations to the transcriptome in the other groups (Figure 3A). The expression of ceramide metabolism genes was investigated from NGS data. C_2_-ceramide treatment upregulated most of the ceramide metabolic genes, but cells cotreated with diTFPP with C_2_-ceramide showed an expression map distinct from that of those treated with C_2_-ceramide only. The results revealed alterations in sphingolipid metabolic genes (Figure 3B,C). Ceramides can be categorized as prosurvival or proapoptotic ceramides. Ceramide and sphingosine are associated with the proapoptotic function that induces endoplasmic reticulum stress, which activates the unfolded protein response and ultimately results in apoptosis [40,41]. On the other hand, the prosurvival class of ceramides usually comprises metabolites of ceramide and sphingosine, such as glucosylceramide (GlcCer), ceramide-1-phosphate (C1P), or sphingosine-1-phosphate (S1P), all of which inhibit caspase activation and induce autophagy to prevent apoptosis and induce cell proliferation [8,18,19,42]. Therefore, the balance of ceramides is critical in the initiation of apoptosis and regulation of cell fate [8]. The expression of genes that catalyze pro-apoptotic ceramides into prosurvival ceramides, such as SPHK1, SPHK2, CERK, and UGCG, was upregulated after C_2_-ceramide treatment (Figure 3B). However, the changes in response to C_2_-ceramide treatment were subverted by cotreatment with diTFPP. In contrast, the expression of the genes ASAH1 and ASAH2, which mediate sphingosine catalysis into ceramide, was upregulated (Figure 3B). These alterations in the expression of ceramide metabolic proteins suggested that the potential mechanism of diTFPP/C_2_-ceramide treatment is apoptosis activation. To investigate the potential role of diTFPP in this process, we analyzed the NGS data of C_2_-ceramide and C_2_-ceramide/diTFPP treatment with gene set enrichment analysis (GSEA) and found that the combined treatment of diTFPP and C_2_-ceramide decreased the gene sets enriched in DNA replication and cell cycle transition pathways (Figure 3D,E), suggesting that diTFPP treatment induces cell cycle arrest.

Additionally, we clustered the NGS data by gene expression level and found 1986 downregulated genes and 2240 upregulated genes (Figure 3F,H). Gene ontology (GO) analysis was utilized to analyze the upregulated and downregulated genes. Downregulated genes were found to be primarily associated with the cell cycle and mitosis, suggesting that cotreatment with diTFPP/C_2_-ceramide results in cell cycle and mitosis arrest (Figure 3G). On the other hand, upregulated genes were found to be associated with protein transport, ROS homeostasis, and autophagy (Figure 3I).

### 3.4. The Autophagy Regulatory Mechanism Plays a Critical Role in diTFPP/C_2_-Ceramide-Induced Apoptosis

Autophagy plays a role in the apoptotic pathway, and the fate of cells—either cell survival or cell death—depends on the regulatory mechanism [43]. From the NGS data, autophagic genes were found to be upregulated, suggesting that autophagy plays an essential role in diTFPP/C_2_-ceramide-induced cell apoptosis. Acidic vesicular organelle (AVO) staining was utilized to detect the generation of autophagosomes to verify the activation of autophagy. The results indicated an increase in AVO signaling after diTFPP/C_2_-ceramide cotreatment (Figure 4A,B). Nevertheless, microtubule-associated protein 1A/1B light chain 3B (LC3B) was modified to LC3B-I and LC3B-II and showed an additive effect after diTFPP or C_2_-ceramide treatment. Additionally, the expression of Beclin-1, a protein associated with autophagosome formation, increased. The alterations in protein expression reflect autophagy activation (Figure 4C). Additionally, we treated the cells with the autophagy inhibitor 3-MA before treatment with C_2_-ceramide and diTFPP and observed that the levels of the apoptotic marker cleaved PARP-1 were significantly decreased in the 3-MA-treated group (Appendix A).

### 3.5. ROS Formation Is Involved in diTFPP/C_2_-Ceramide-Induced Cell Death

The NGS data also demonstrated that genes related to the response to H_2_O_2_ and cell redox homeostasis were upregulated, suggesting that ROS play a role in diTFPP/C_2_-ceramide treatment. ROS formation in the cell was measured with dihydroethidium (DHE), which is a superoxide-targeting red fluorescence reagent. The results indicated that combined treatment with diTFPP and C_2_-ceramide significantly induced the formation of ROS (Figure 5A,B). Additionally, ROS-associated proteins such as catalase were found to increase after diTFPP/C_2_-ceramide treatment. The superoxide dismutase 1 (SOD1)-to-superoxide dismutase 2 (SOD2) "switch" phenomenon occurred, which was also observed in our previous study and was associated with autophagic apoptosis (Figure 5C). Mitochondria are the major source of ROS production [44]. We observed the generation of mitochondrial ROS with MitoSOX red and found that the C_2_-ceramide/diTFPP combined treatment induced considerable mitochondrial ROS production in HA22T cells (Appendix A). In addition, the ROS inhibitor NAC was utilized to inhibit the formation of ROS to observe the effect of ROS on C_2_-ceramide/diTFPP-induced apoptosis. The results indicated that pretreatment with NAC blocked the formation of cleaved PARP-1 and consequently inhibited apoptosis (Appendix A).

## 4. Discussion

HCC is a severe disease that was responsible for over 700,000 deaths in 2018 [1,2]. The results presented in this study indicated the apoptotic role of C_2_-ceramide and diTFPP cotreatment in the HA22T HCC cell line. The cells treated with C_2_-ceramide exhibited pyknosis, which is a cellular process that reduces cellular and nuclear volume and is considered a characteristic of apoptosis or necrosis [45,46]. Although both cell death processes can involve pyknosis, annexin V/PI staining and the expression of apoptotic proteins revealed the apoptotic role of C_2_-ceramide/diTFPP treatment.

The assessment of sphingolipid metabolism provided evidence that the disruption of ceramide metabolism induces cell apoptosis or survival. Sphingolipid metabolites show two different mechanisms: some induce cell apoptosis, while others promote cell survival [40]. Treatment with C_2_-ceramide upregulated most of the ceramide metabolic genes, catalyzing apoptotic ceramide into glycosylceramide, ceramide-1-phosphate, and sphingosine-1-phosphate, which are known for cell survival (Figure 3B), to maintain healthy homeostasis from exogenous ceramide administration. UGCG is a glucosylceramide synthase that catalyzes the formation of glucosylceramide from ceramide [47]. UGCG has been reported to be a regulator of Akt activation and a promoter of cell proliferation [48]. Salustiano and Previato also discovered that UGCG is involved in multidrug resistance [49]. The synthesis of ceramide-1-phosphate (C1P) and sphingosine-1-phosphate (S1P) is associated with ceramide kinase (CERK) and sphingosine kinase (SPHK), which phosphorylate ceramide and sphingosine, respectively [50]. C1P and S1P play a role in cell survival by regulating members of the Bcl family to inhibit apoptosis [51]. Sphingosine and ceramide are sphingolipids with pro-apoptotic functions [40]. Akt, c-Myc, and Bcr-Abl are oncogenes that have been reported to be suppressed by the regulatory mechanisms of ceramide and sphingosine synthesis [52,53,54]. Ceramide has also been reported as a necessary mediator of caspase-3 cleavage in response to radiation [55]. The disruption of C_2_-ceramide signaling indicated the potential pathway by which diTFPP induces apoptosis.

In this study, ROS played a vital role in diTFPP/C_2_-ceramide-induced cell death. According to the alteration of sphingolipid metabolism, we assumed that ROS production is associated with sphingolipid disturbance. In our previous study, C_8_-ceramide induced lung cancer apoptosis by producing ROS and disrupting superoxide dismutase (SOD) expression [11]. Knupp and Chang indicated that mutant-induced sphingolipid accumulation led to mitochondrial dysfunction and ROS production [56]. The study provided a view of sphingolipid-induced ROS production associated with the dysregulation of mitochondria. In an early study, exogenous ceramide was found to inhibit the activity of mitochondrial respiratory chain (MRC) proteins and induce the production of ROS [57,58,59]. The NGS data also showed that mitochondrial (mt)-proteins were all upregulated after diTFPP/C_2_-ceramide treatment (data not shown). The alteration of mt-protein expression might compensate for MRC inhibition by ceramide accumulation.

Autophagy also plays a vital role in apoptosis, regulating the fate of cells from survival to death [60]. The crosstalk of apoptosis and autophagy has been reported to be associated with sphingolipid metabolism [15]. As mentioned before, ceramide inhibits Akt protein kinase via pyrophosphatase protein 2A (PP2A), resulting in the activation of mammalian target of rapamycin (mTOR)-mediated autophagy [61,62]. Ceramide is also associated with the transcription and lipidation of LC3, an autophagic protein associated with autophagosome generation during lipidation [63]. Sun and Zhu reported that ceramide treatment mediated JNK activation and the phosphorylation of c-Jun and was associated with the transcription and lipidation of LC3 [63]. We also observed the upregulation of endoplasmic reticulum (ER) stress response proteins, indicating that ceramide-mediated ER stress regulates the activation of autophagy [15] (data not shown). These results indicate that diTFPP/C_2_-ceramide-mediated cell death is multifactorial and is associated with autophagy, mitochondrial dysfunction, ROS production, and ER stress-mediated responses.

## 5. Conclusions

The results indicated that diTFPP disrupts C_2_-ceramide metabolism, leading to the activation of autophagy, which results in the formation of ROS and ultimately induces cell apoptosis in HA22T hepatocellular carcinoma (Figure 6).

## Figures and Tables

**Figure 1 antioxidants-10-00394-f001:**
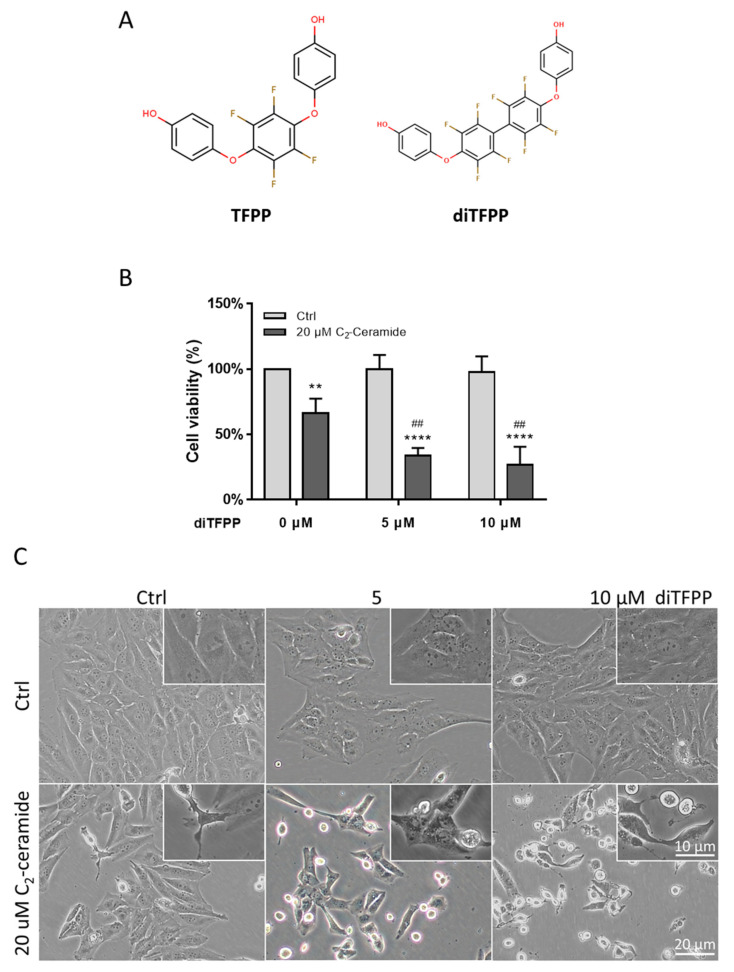
diTFPP and C_2_-ceramide cotreatment induced cell death of HCC cells. (**A**) The structure of phenoxy phenol compounds, TFPP, and diTFPP. (**B**) Cell viability of HA22T cells after diTFPP and/or C_2_-ceramide treatment for 24 h. n = 3, ** *p* < 0.01, **** *p* < 0.0001 compared with the control and 0 μM diTFPP groups. ## *p* < 0.01 compared with the C_2_-ceramide with no diTFPP treatment group; all data are presented as the mean ±SD of three independent experiments. (**C**) HA22T cell morphology after C_2_-ceramide and/or diTFPP treatment for 24 h. Magnification: 100×.

**Figure 2 antioxidants-10-00394-f002:**
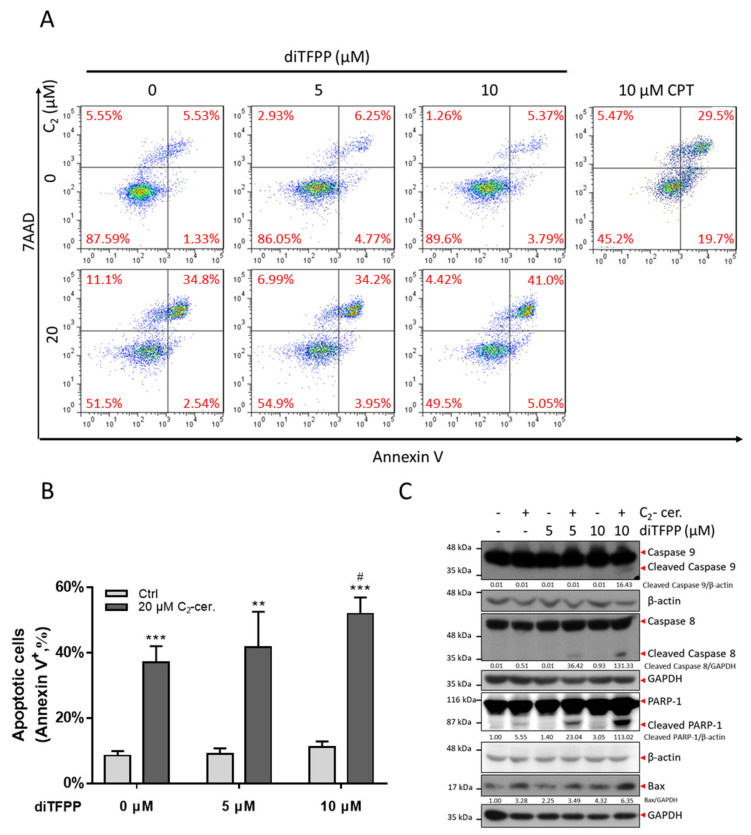
diTFPP promotes C_2_-ceramide-mediated cell death via apoptosis. (**A**) Annexin V/7AAD double staining indicated cell death in response to 24 h of diTFPP/C_2_-ceramide treatment. Treatment with 10 μM CPT was considered the positive control. (**B**) Quantification of (A). n = 3, ** *p* < 0.01, *** *p* < 0.001 compared with the control; # *p* < 0.05 compared with the C_2_-ceramide only group. (**C**) Western blot analysis of apoptotic protein expression after diTFPP and C_2_-ceramide (C_2_-cer.) treatment. Cleaved caspase-9 and Bax expression indicated intrinsic apoptosis, cleaved caspase-8 indicated extrinsic apoptosis, and cleaved PARP-1 indicated the hallmark of apoptosis. GAPDH and β-actin as internal controls. The fold changes of cleaved caspases were normalized with their internal control.

**Figure 3 antioxidants-10-00394-f003:**
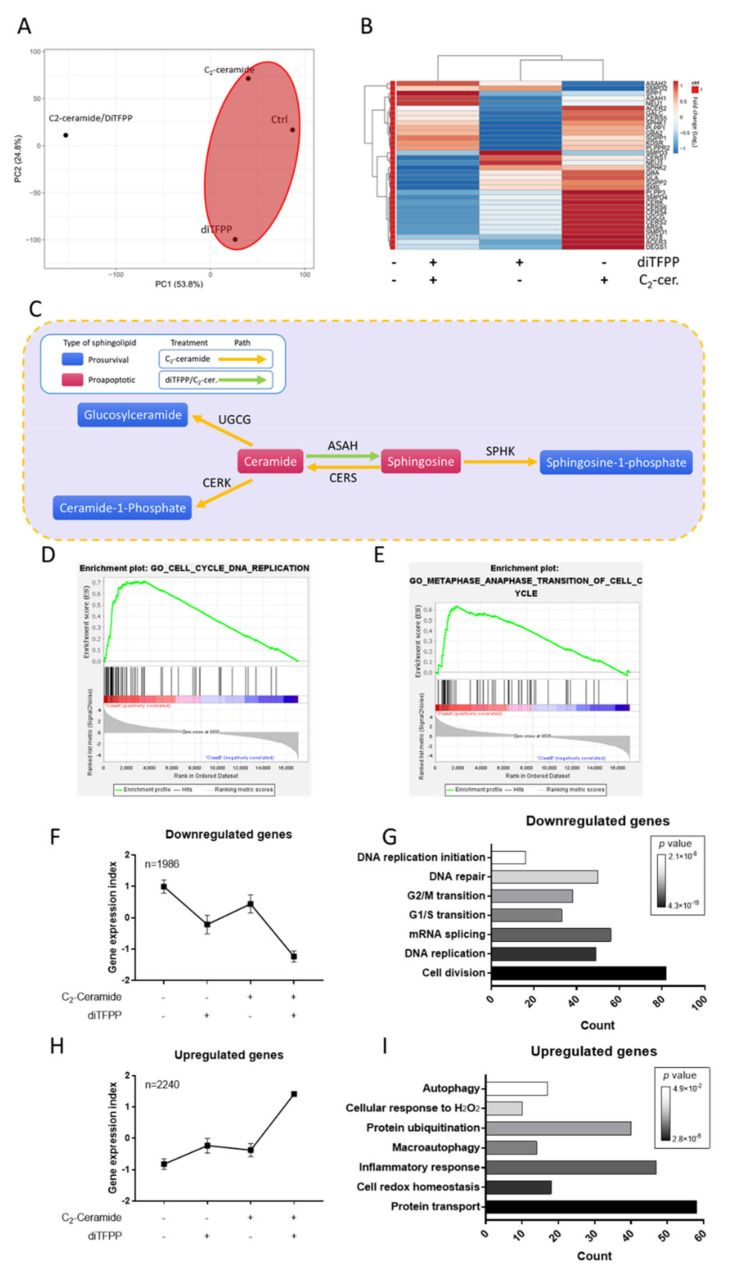
NGS analysis of diTFPP/C_2_-ceramide treatment indicating the alteration of sphingolipid metabolism and the potential regulatory mechanisms. (**A**) PCA of the four treatment groups. (**B**) The expression of sphingolipid metabolic genes after 24 h of 10 μM diTFPP and/or 20 μM C2-ceramide (C_2_-cer.) treatment. (**C**) Schematic diagram of sphingolipid metabolic alteration. (**D** and **E**) GSEA of differences in the biological processes (BP) between C_2_-ceramide (Class A)- and C_2_-ceramide/diTFPP (Class B)-treated cells. (**F**) Alterations in the expression pattern of downregulated genes after diTFPP/C2-ceramide treatment, with average expression set as 0 and the standard deviation set as 1. (**G**) GO term analysis of downregulated genes after diTFPP/C2-ceramide treatment. (**H**) The expression alteration of upregulated genes after diTFPP/C2-ceramide treatment, with average expression set as 0 and the standard deviation set as 1. (**I**) GO term analysis of upregulated genes after diTFPP/C2-ceramide treatment.

**Figure 4 antioxidants-10-00394-f004:**
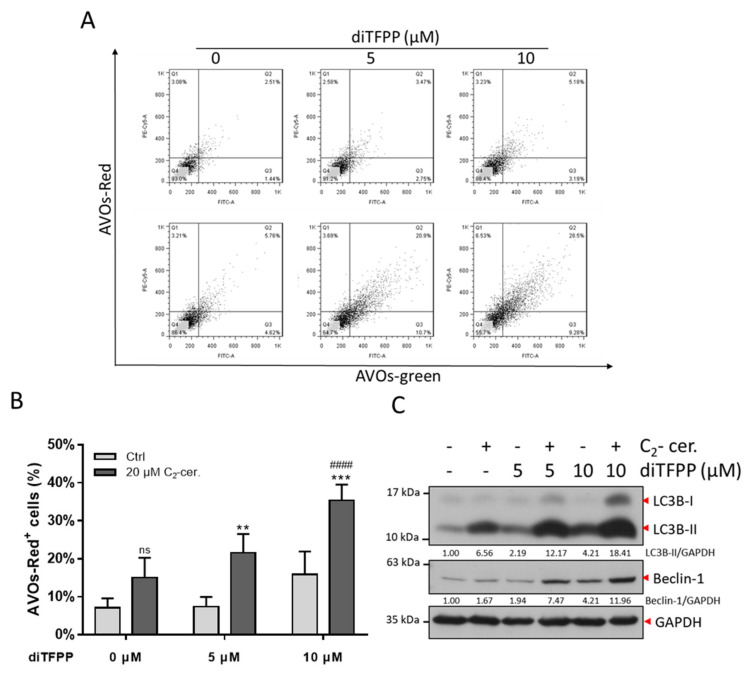
Autophagy is involved in diTFPP/C_2-_ceramide treatment. (**A**) AO staining after 24 h of diTFPP/C_2-_ceramide treatment. The x-axis indicates green fluorescence, and the y-axis indicates red fluorescence (AVO-positive cells). (**B**) Quantification of (A). ns, no significant *p*>0.05, ** *p* < 0.01, *** *p* < 0.001 compared with the control groups. #### *p* < 0.0001 compared with the C_2_-ceramide alone group; all data are presented as the means ±SD of four independent experiments. (**C**) Western blot analysis of autophagic proteins after 24 h of diTFPP/C_2-_ceramide treatment. GAPDH as an internal control.

**Figure 5 antioxidants-10-00394-f005:**
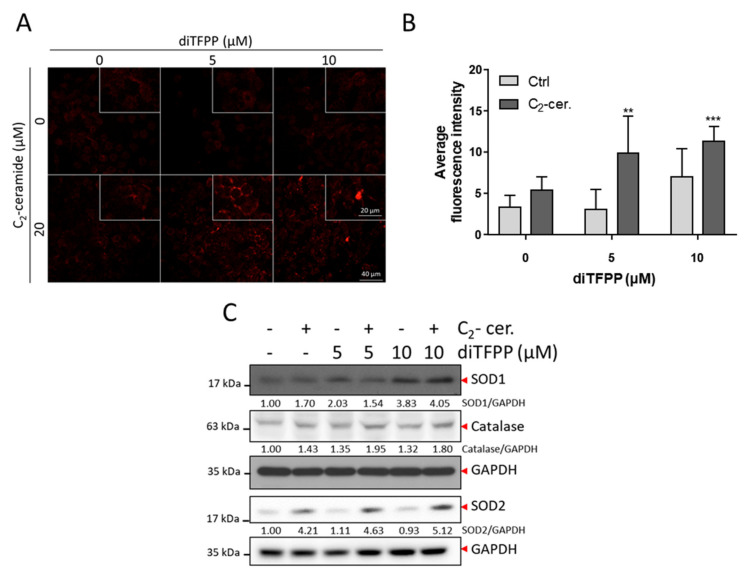
ROS plays a role in diTFPP/C_2_-ceramide-mediated cell death. (**A**) DHE staining with red fluorescence indicated ROS production in diTFPP/C_2_-ceramide-treated HA22T cells. (**B**) The quantification of (A). ** *p* < 0.01, *** *p* < 0.001 compared with the control groups. (**C**) Western blot analysis of ROS metabolic protein expression after diTFPP/C_2_-ceramide treatment. GAPDH as an internal control.

**Figure 6 antioxidants-10-00394-f006:**
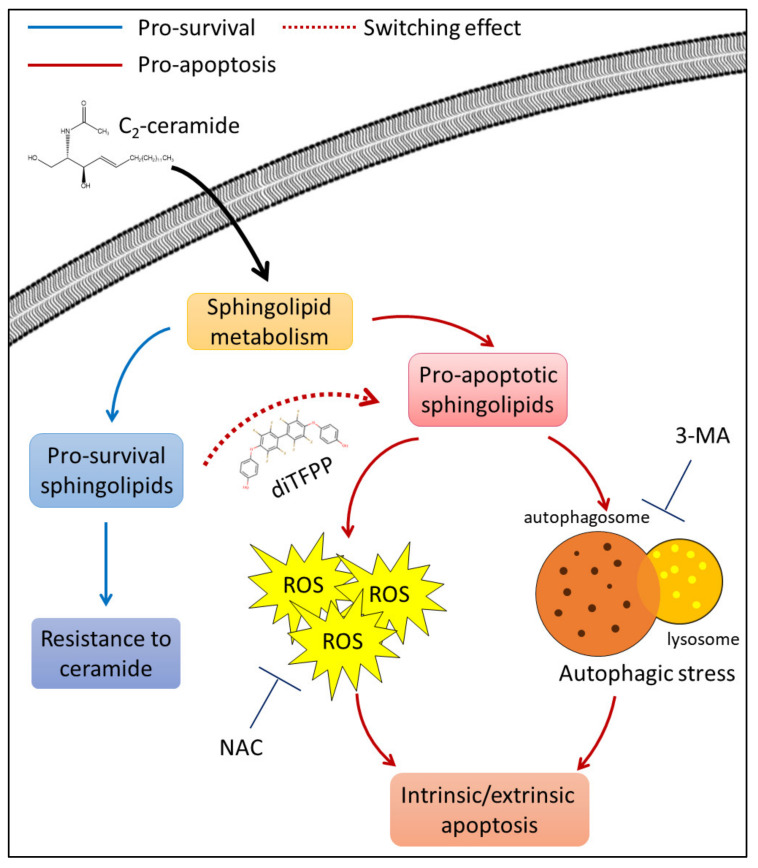
The potential regulatory mechanism of diTFPP in C_2_-ceramide-mediated cell death. Endogenous sphingolipid metabolism favors the conversion of exogenous C_2_-ceramide to prosurvival sphingolipids and causes the attenuation of C_2_-ceramide-induced anti-HCC effects, including anti-proliferation and apoptosis induction in HCC cells. In contrast, cotreatment with C_2_-ceramide and diTFPP promotes the switching of sphingolipid metabolism from prosurvival to proapoptosis, enhances ROS production and autophagic stress, and eventually sensitizes HCC cells to C_2_-ceramide-induced apoptosis.

## Data Availability

The authors confirm that the data supporting the findings of this study are available within the article.

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
