# Peer review of "The Phenoxyphenol Compound diTFPP Mediates Exogenous C_2_-Ceramide Metabolism, Inducing Cell Apoptosis Accompanied by ROS Formation and Autophagy in Hepatocellular Carcinoma Cells"

_antioxidants, 2021, doi:10.3390/antiox10030394_

Round 1

Reviewer 1 Report

The authors have made significant changes since their previous submission.

Most of the requests have been performed, including new controls, better images and more detailed characterization of the involvement of caspase-8.

The only minor question that I have is that in cases there are 2 bands in the WB (caspase-9/cleaved caspase-9 for example) it is not clear to what the number below refers: ratio of caspase-9? Ratio of cleaved caspase-9?

Author Response

The authors have made significant changes since their previous submission.

Most of the requests have been performed, including new controls, better images and more detailed characterization of the involvement of caspase-8.

The only minor question that I have is that in cases there are 2 bands in the WB (caspase-9/cleaved caspase-9 for example) it is not clear to what the number below refers: ratio of caspase-9? Ratio of cleaved caspase-9?

Response: We thank the reviewer’s accreditation. According to the caspase feature, passing the apoptosis signaling while activating, we measured the fold change of cleaved caspase-9 which was normalized with internal control (the ratio of cleaved caspase-9 to β-actin the internal control). We also added the above description to the revision.

Figure 2. diTFPP promotes C2-ceramide-mediated cell death via apoptosis. (A) Annexin V/7AAD double staining indicated cell death in response to 24 hours of diTFPP/C­2-ceramide treatment. Treatment with 10 μM CPT was considered the positive control. (B) Quantification of (A). n=3, **p<0.01, ***p<0.001 compared with the control; #p<0.05 compared with the C2-ceramide only group. (C) Western blot analysis of apoptotic protein expression after diTFPP and C­2-ceramide (C2-cer.) treatment. Cleaved caspase-9 and Bax expression indicated intrinsic apoptosis, cleaved caspase-8 indicated extrinsic apoptosis, and cleaved PARP-1 indicated the hallmark of apoptosis. GAPDH and β-actin as internal controls. The fold changes of cleaved caspases were normalized with their internal control.

Reviewer 2 Report

Dear Dr. Laura Soto Hinojosa
Special Issue Editor of Antioxidant

I read the revised version of manuscript titled "The phenoxyphenol compound diTFPP mediates exogenous C2-ceramide metabolism, inducing cell apoptosis accompanied by ROS formation and autophagy in hepatocellular carcinoma cells".

I appreciated the efforts made by authors to improve the manuscript quality. I think that, in this form, the manuscript is suitable for publication on Antioxidant Journal

Regards

Author Response

#Reviewer 2

I read the revised version of manuscript titled "The phenoxyphenol compound diTFPP mediates exogenous C2-ceramide metabolism, inducing cell apoptosis accompanied by ROS formation and autophagy in hepatocellular carcinoma cells".

I appreciated the efforts made by authors to improve the manuscript quality. I think that, in this form, the manuscript is suitable for publication on Antioxidant Journal

Response: We thank the reviewer’s accreditation and the encouragement.

This manuscript is a resubmission of an earlier submission. The following is a list of the peer review reports and author responses from that submission.

Round 1

Reviewer 1 Report

Dear Dr. Laura Soto Hinojosa, assistant editor of Antioxidant

I read the revised version of manuscript titled "The phenoxyphenol compound diTFPP mediates exogenous C2-ceramide metabolism, inducing cell apoptosis accompanied by ROS formation and autophagy".

I appreaciated the efforts made by the authors to improve the quality of manuscrit. I think that the revised version of manuscript is suitable for publication on Antioxidant

Regards

Response: We thank the reviewer for the accreditation and encouragement

Reviewer 2 Report

This is a well-designed study that addresses a topic of interest to the readers of Antioxidants.

However, the following issues must be addresed:

- Fig. 2A – Image should be provided with higher resolution and percentages in each quadrant must be made clear, which is not the case currently.

Response: We thank the reviewer for the suggestions, and we updated the data with a higher resolution figure in the revision.

- If the authors wish to valorize the finding of cleaved caspase-8 as an indication of extrinsic pathway, they must provide data for tBid as well.

Response: Because an antibody against tBid is not available in our laboratory presently, alternatively we treated the cells with Z-IETD-FMK, a caspase 8 inhibitor, and examined cell viability to verify caspase 8 activation. As a result,
the treatment reversed the cell death induced by C2-ceramide/diTFPP cotreatment (please see the figure below).

- Densitometric analysis for all proteins in the western blot must be provided in order to discuss them even if from a semi-quantitative point of view.

Response: We added semiquantitative data to all western blot data in the revised manuscript.

Figure 6 is of low resolution

Response: We updated Figure 6 with a high-resolution illustration.

- A positive control for apoptosis experiments must be provided.

Response: We added the positive control (10 μM CPT) to the data in the revision

Minor issues:

Line 37: "The phenolic phenol compound (...)" is an incorrect designation.

Response: We thank the reviewer for pointing out the mistake. We corrected the mistake in the revision.

In light of this, I advise for major revisions.